# Molecular Pathophysiology and Potential Therapeutic Strategies of Ketamine-Related Cystitis

**DOI:** 10.3390/biology11040502

**Published:** 2022-03-24

**Authors:** Chin-Li Chen, Sheng-Tang Wu, Tai-Lung Cha, Guang-Huan Sun, En Meng

**Affiliations:** 1Division of Urology, Department of Surgery, Tri-Service General Hospital, National Defense Medical Center, Cheng-Gung Road, Taipei 114, Taiwan; iloveyou@mail.ndmctsgh.edu.tw (C.-L.C.); doc20283@mail.ndmctsgh.edu.tw (S.-T.W.); tailung@mail.ndmctsgh.edu.tw (T.-L.C.); ghsun@mail.ndmctsgh.edu.tw (G.-H.S.); 2Department and Graduate Institute of Biochemistry, National Defense Medical Center, Cheng-Gung Road, Taipei 114, Taiwan

**Keywords:** ketamine-related cystitis, lower urinary tract symptoms, molecular pathophysiology

## Abstract

**Simple Summary:**

Ketamine, an *N*-Methyl-D-aspartate receptor antagonist, acts as a quick-acting and general anesthetic for human and animal anesthesia. In recent years, ketamine has been widely abused among young people and commonly abused in combination with other drugs. Continued use of ketamine will damage mental status and cognitive function. Some abusers will suffer from ketamine-related cystitis (KC) and experienced severe lower urinary tract symptoms, such as frequent urination, hematuria, urinary incontinency, and bladder pain. The etiology of KC is still uncertain and therefore no definite treatment has been established. This review article focuses on current evidence of molecular pathophysiology and possible therapies for KC.

**Abstract:**

Ketamine was first synthesized as a clinical medicine for anesthesia in 1970. It has been used as a recreational drug because of its low cost and hallucination effect in the past decade. Part of ketamine abusers may experience ketamine-related cystitis (KC) and suffer from lower urinary tract symptoms, including urinary frequency, urgency, and severe bladder pain. As the disease progression, a contracted bladder, petechial hemorrhage of the bladder mucosa, and ureteral stricture with hydronephrosis may occur. The pathophysiology of KC is still uncertain, although several hypotheses have been raised. Cessation of ketamine abuse is critical for the management of KC to prevent progressive disease, and effective treatment has not been established. Research has provided some theoretical bases for developing in vitro experiments, animal models, and clinical trials. This review summarized evidence of molecular mechanisms of KC and potential treatment strategies for KC. Further basic and clinical studies will help us better understand the mechanism and develop an effective treatment for KC.

## 1. Introduction

Ketamine, a non-competitive N-methyl-D-aspartic acid receptor complex antagonist, has acted as a short-acting general anesthesia agent or an analgesic since the 1960s [1]. Ketamine has been used as a recreational drug and increasing abuse in nightclubs since the 1980s. It may result in hallucinations and stimulates out-of-body experiences [2].

As most of the drug abusers are multiple drug addicts, the actual prevalence of ketamine use is difficult to calculate. Pal et al. reported that 18.7% of drug abusers have ketamine abuse for a lifetime in the United States, and 5.8% of drug abusers have had ketamine abuse recently [3]. Ketamine-related cystitis (KC) was one of the severe complications in ketamine abusers and first reported in a series of 10 young Chinese patients in 2007 [4]. The incidence rate of KC in long-term ketamine users was as high as 30% [5,6].

Ketamine abusers with KC suffer from various lower urinary tract symptoms (LUTS), such as frequent urination, urgent urination, nocturia, urinary incontinence, hematuria, painful micturition, and lower abdominal pain [7,8,9]. Additionally, the bladders with KC showed thickening, contracted, and inflammation [4,10].

In a survey of ketamine users, 26.6% of recent ketamine abusers presented with LUTS, and these problems were significantly correlated with both dose-related and frequency-related ketamine abuse [5]. The LUTS improved after stopping the use of ketamine. However, continuous usage of ketamine will worsen LUTS and further cause pathological changes of the urinary tract [11].

How ketamine affects the bladder and causes the damage of urothelial cells remains unclear. It has been proposed that ketamine and its active metabolite stay in the urine of the bladder and cause direct toxic effects on the urothelial cells of the bladder [12]. Several proposed etiologies have also been raised, including activation of inflammatory cells, dysfunction of bladder barrier, dysregulation of neurotransmission, abnormality of cell apoptosis, and overexpression of oxidative stress. This article reviews the current medical evidence of molecular and pathological mechanisms of KC.

## 2. Clinical and Pathological Features of Ketamine-Related Cystitis

The typical symptoms of KC include severe dysuria, urinary frequency with a minimal amount of urine, urgency, urge incontinence, hematuria, and postmicturition pelvic pain [10]. The incident rate of LUTS in ketamine abusers was as high as 49.5% [13]. Regarding suffering from LUTS between male and female ketamine abusers, there was no significant difference [5].

Cystoscopy showed varying degrees of mucosal inflammation, multiple edematous changes of the mucosa, and neovascularization in the bladder. Glomerulation, petechial hemorrhages, and ulceration of the urinary bladder may also be found in severe patients [14]. The urodynamic study demonstrated significantly decreased capacity of urinary bladder (possibly as decreased as 10–20 mL per void) and hyperactivity of detrusor muscles that might lead to urge incontinency with diaper dependence [15]. Persistent ketamine abuse can also result in progressive complications involving the upper urinary tract, including stricture of ureter, hydronephrosis, and impairment of kidney function [16].

In microscopical morphology with hematoxylin and eosin (H&E) stain of bladder mucosa with KC, the pathological features showed denuded urothelium, ulceration and hemorrhage (Figure 1A,C) [17]. The KC bladder mucosa may infiltrate with eosinophils, lymphocytes, and plasma cells as an inflammatory reaction. Intravascular accumulation of neutrophils and eosinophils was also noticed (Figure 1B) [18]. The stroma of KC urinary bladder presented with increased deposition of collagen, degeneration of smooth muscles, increased calcification, and vascular proliferation (Figure 1D) [18].

## 3. Molecular Evidence of the Pathophysiology and Potential Biomarkers

Although ketamine may have systemic effect on the urinary tract, direct exposure of ketamine and/or its active metabolite via the urine is more likely to cause destruction of the urothelium and sequential insults [19,20]. Possible pathophysiological mechanisms of KC, including activation of inflammatory cells [21], dysfunction of bladder-urothelial barrier [22], dysregulation of neurotransmission [23], cell apoptosis [24,25], and oxidative stress [26,27], have been proposed. An overview of proposed molecular mechanisms underlies the interaction between ketamine and the bladder (Figure 2). However, the actual mechanism of KC is still uncertain. Previous research has used human bladder tissue, animal models, and urothelial cell lines to explore the etiology of KC. We reviewed current molecular evidence for the etiology of KC as below (Table 1).

### 3.1. Evidence from the Human Bladder Tissue

#### 3.1.1. Inflammatory and Immune Reactivity in Human KC Bladder

Ketamine and its metabolites can affect urothelial cells by direct toxic damage on the bladder of KC, which was first reported in 2007 [10]. As ketamine and norketamine, a metabolite of ketamine, are excreted from the kidney and retained in the urinary bladder, they can interact with the bladder urothelium and induce an inflammatory reaction [28]. An inflammation occurs throughout the bladder stroma with a lot of mast cells and lymphocytes, but predominates with eosinophils in the KC [10].

It has been supposed that ketamine abusers with KC excreted ketamine and its metabolite into the bladder and resulted in immunological hypersensitivity reaction due to interact with the urothelial cells [28]. Elevated serum immunoglobulin IgE was reported as an indicator for the immunological hypersensitivity reaction [41].

Jhang et al. showed the presenting level of serum IgE was significantly higher in KC group compared with the patients having acute bacterial cystitis, interstitial cystitis/bladder pain syndrome (IC/BPS), and control group (33.4 IU/mL). Additionally, serum IgE significantly correlated with VAS. The median VAS pain score in KC patients with serum IgE > 200 IU/mL was significantly higher compared to KC patients with serum IgE < 200 IU/mL. The level of serum IgE showed downregulation of presentation after cessation of ketamine use [21].

Elevated serum immunoglobulin IgE in KC seems to play an essential factor in pathophysiological reactions of bladder hypersensitivity.

Fan et al. reported that ketamine might lead to acute inflammation through the TH1 pathway and provoke generation of IL-6 as a secondary mediator, which was correlated with the dynamic balance of TH17/TREG cell effector functions. The effect of IL-6 can suppress TREG cells and subsequently cause prolonged activation of TH1 and TH2 cells. This may result in damage of tissue and inflammation by immune response. Furthermore, the IL-6 may increase the production of endogenous IL-4 and enhance the formation of IgE in the patients with KC, and subsequently stimulate the differentiation of TH2 cells. This chronic inflammation is associated with TH2-mediated response. Subsequently, it will damage the healing of urothelial cells in the patients with KC [18].

#### 3.1.2. Ketamine Might Affect Neuronal Growth

Brain-derived neurotrophic factor (BDNF) can act on the neural system including central and peripheral nervous system to maintain the survival of neurons and promote neuronal growth, development, and differentiation [42,43]. BDNF plays several critical functions, including learning, memory, synaptic plasticity, and cognitive function [44].

Ketamine can affect the functions of the central nervous system and peripheral nerves. In recent publications, ketamine use has been to shown to alter BDNF in the human study as well as in the animal study. Compared with ordinary people, there is around twice the serum concentration of BDNF in chronic ketamine abusers [45]. In contrast, Ke et al. reported an opposite result of serum BDNF in KC patients [46]. They showed a significantly lower level of serum BDNF in chronic abusers of ketamine than among healthy people. Due to such inconsistent results, it needs further studies to confirm the importance of BDNF on ketamine abusers.

Baker et al. reported that neurogenesis was found in the stroma of patients with KC [23]. Their bladders were infiltrated with fine neurofilament protein (NFP^+^) nerve fibers. Hyperplasia of the peripheral nerve fascicles labeled with nerve growth factor receptor (NGFR^+^) of the lamina propria cells was also identified in their bladders. Superficial neuroma-like lesions in the bladders of KC patients were likely to cause severe pain. These findings demonstrated the hyperplastic and reactive response in KC patients’ bladders. However, the role of ketamine on the neuronal network of the bladder is yet to be established.

#### 3.1.3. Ion Channels Might Be the Mechanosensory in the KC Bladder

Urothelium serves as a barrier to protect the bladder stroma and a sensory receptor to manage the contractile activity of bladder. Yang et al. reported that some ion channels, such as transient receptor potential (TRP) channels, are possible sensor molecules for bladder contraction [33]. It presented with significantly higher expression level of TRP cation channel subfamily V member 1 (TRPV1) and TRPV4 in the KC bladder of abused patients than control patients. The expression of TRPV1 was inversely correlated with maximal bladder capacity. The expression of TRPV4 was positively correlated with the velocity to reach maximal pressure of the detrusor. It suggested that elevation of TRPV1 and TRPV4 in the bladder of KC was correlated with smaller capacity and stronger contractility of bladder.

### 3.2. Evidence from the Animal Model

KC is a new disease entity and rare in the general population. It is difficult to determine and conduct clinical research on these unusual conditions. Animal models are critical for the future development of therapeutics. Rodent models have been widely used for KC research. Because of different physiological responses to the action of ketamine, differences in experimental methods, and lack of adequately defined experimental parameters, rodent models of KC might have restricted translational value. Nevertheless, there is a considerable utility for rodent models in screening drugs for treating KC [47].

#### 3.2.1. Deficiency of Urothelial Junction-Associated Protein Impaired the Barrier

Disruption of the urothelial barrier can lead to leakage of irritative stimulants and urinary potassium to the suburothelium and cause an inflammatory reaction and provoke cystitis.

The glycosaminoglycan (GAG) layer is a protective mucosa composed of glycoproteins and proteoglycans, which located on the bladder urothelium. Yeh et al. demonstrated that rats treated with intravesical instillation of ketamine presented with damaged GAG layer in the bladder [48].

Zonula occludens-1 (ZO-1), a cytoplasmic protein, constitutes the scaffold that connects the tight junction-associated transmembrane proteins to the perijunctional actin cytoskeleton in the urothelium and regulates adjacent cells with transducing signals [49]. Uroplakins (UPKs), the protein components of uroplaques, are urothelial transmembrane proteins, including UPIa, UPIb, UPII, and UPIII [50]. The expression level of ZO-1 protein was decreased in the bladder of IC/BPS patients as well as the patients with chronic spinal cord injury [51]. A significant decrease of ZO-1 protein was also reported in the rat bladder treated by ketamine [22]. Lee et al. reported that ketamine-treated rats presented with a significant decrease in urothelial tight junction proteins (claudin-4 and ZO-1), adhesion protein (E-cadherin), and urothelial umbrella cells (UPIII) [32]. Urothelial junction-associated proteins damaged after ketamine treatment and subsequently resulted in a defect of the urothelial-lining layer in the bladder. Decreased expression of UPIII was also found in the bladder of KC patients [52].

#### 3.2.2. Oxidative Stress Species Enhanced Bladder Hyperactivity

Reactive oxygen species (ROS) can be found in the pathological status of the urinary bladder, such as bladder outlet obstruction [53,54], ischemia reperfusion [55,56], and inflammation. Inflammation-associated ROS may lead to a hyperactive bladder [57,58]. Previous research also suggested that oxidative stress that occurred in the bladder could also damage the function of muscarinic receptors and reduce detrusor contractility [59,60]. Liu et al. demonstrated that in increases in the mitochondria-dependent pathway, as well as endoplasmic reticulum-dependent pathway, wounds upregulated the generation of oxidative stress in rat bladders after one month of ketamine treatment [26]. In contrast, mRNA expressions of the antioxidant enzymes Mn-SOD (SOD2), Cu/Zn-SOD (SOD1), and catalase significantly decreased. These results suggested that ROS involves apoptosis of bladder cells and defects of urothelium barriers. The oxidative stress mechanism is supposed to be a critical factor of bladder overactivity and ulceration in KC [26].

#### 3.2.3. Ketamine Increased Purinergic Neurotransmission Caused Detrusor Overactivity

P2X1 purinergic receptors, which are located on the detrusor muscles, play an active role in mediating bladder contraction in mice [29]. Meng et al. presented an increased expression of P2X1 purinergic receptor immunoreactivity in the bladder muscle layer of mice after ketamine treatment compared to the control group at 8 weeks [36]. However, it showed no significant difference of the immunoreactivity of the M2- and M3-muscarinic acetylcholine receptors (mAChRs) between the ketamine-treated group and the control group at 8 weeks. The contraction of detrusor muscle evoked by adenosine triphosphate increased in ketamine-treated group. The result of upregulated P2X1 receptor expression underlies detrusor overactivity in KC and subsequently induces bladder dysfunction [36].

#### 3.2.4. Ketamine Moderate Ion Channels in the Bladder Smooth Muscle and Affect the Bladder Function

Ketamine-related cystitis may cause bladder dysfunction, such as decrease of voiding pressure and voiding interval, poor bladder compliance, and increased bladder pain after intravesical infusion of ketamine. The L-type calcium channel (Cav1.2) plays a crucial factor in detrusor muscle contractility of the urinary bladder [61]. Chen et al. demonstrated that ketamine inhibits calcium influx and contraction of smooth muscles by inhibiting Cav1.2. channels in the rat bladder. Additionally, ketamine inhibits genes expression and transcription factors which induction by Cav1.2 [15]. As a result, ketamine can prevent activation of Cav1.2 in the smooth muscle of KC bladder, and subsequently leads to voiding dysfunction.

Shen et al. performed microarray analysis for the ketamine-treated mouse bladders and found that the expression of two genes, KCNMA1 (potassium calcium-activated channel subfamily M α 1) and KCNMB4 (potassium calcium-activated channel subfamily M regulatory β subunit 4), was downregulated. These two genes are involved in the calcium signaling pathway and constitute the BK channels, which were large-conductance voltage- and Ca2^+^-activated channels, with the α and β subunits [37,38]. BK channels may regulate the frequency of Ca2^+^ sparks to contract the urinary bladder smooth muscle [62]. As a result, downregulation of these two genes, KCNMA1 and KCNMB4, can lead to a decreased presentation of BK channels and subsequently cause dysfunction of bladder contraction and micturition [37,38].

#### 3.2.5. Change of Extracellular Matrix Gene Expression May Involve Bladder Fibrosis

Collagen deposition in the submucosal layer (fibrosis) and detachment of urothelial cells were often presented in the ketamine treated bladder of mice [36]. Matrix deposition and interstitial fibrosis of the ketamine-treated bladder in rats showed upregulated levels of fibrosis-related genes, including collagen I (COL I), collagen III (COL III), fibronectin, and transforming growth factor-β (TGF-β) [39].

From global gene expression microarray analysis of the animal’s bladder tissues, ketamine treatment induced upregulation of 10 genes and downregulation of 36 genes [25]. In total, 52% of keratin family genes presented with downregulation. Among these keratin genes, the top three downregulated genes were keratin 6 a, 13, and 14 [25]. It was supposed that cytotoxicity and downregulation of the keratin gene were important factors of KC [25].

A previous study used microarray analysis to determine the gene network of KC to clarify its development. As compared with the control group, they reported 284 upregulated and 527 downregulated genes of mice after ketamine treatment. There were nine pathways correlated with KC, including matrisome (ECM (extracellular matrix) glycoproteins, matrisome and matrisome associated), calcium signaling pathway, small cell lung cancer, MAPK (mitogen-activated protein kinase) signaling, regulation of actin cytoskeleton, neuroactive ligand receptor interaction, and complement and coagulation cascades [37].

Upregulated genes of ECM included FN1 (fibronectin 1), fibulin 2, fibrinogen-like 2, laminin γ2 (LAMC2), and collagen type 1 α 2 (COL1A2). Other upregulated genes related to ECM involved versican (VCAN), angiotensinogen (AGT), and C-type lectin domain family 4 member D [37]. After ketamine treatment, the fibrosis of the bladder in KC mice may result from ECM modulators produced by active fibroblasts [37]. ECM-associated genes were considered to serve an important role in bladder fibrosis [30].

In summary, according to these gene expression profile studies and gene network analysis, long-term ketamine abuse may provoke bladder fibrosis due to upregulation of submucosal collagen deposition and downregulation of keratin production.

#### 3.2.6. Immune and Inflammatory Signaling Pathways Altered in KC Bladders

Methoxetamine, an antagonist of N-methyl-D-aspartate receptor that serves as synthetic psychoactive ketamine analog, was also used for recreational purposes [63]. Both ketamine and methoxetamine treated rats significantly increased pro-inflammatory cytokines and chemokines, including IL-1β, IL-6, CCL-2, CXCL-1, CXCL-10, NGF, and cyclooxygenase-2 (COX-2). These results indicate that both ketamine and methoxetamine use can result in bladder inflammation in rats [39].

Juan et al. found that ketamine and norketamine induced increased COX-2 expression and accelerated nuclear factor-κB (NF-kB) translocation in the rat bladder’s urothelial cells. They also presented that COX-2 inhibitor prevents the intensity of interstitial fibrosis caused by ketamine. This study concluded that the expression of COX-2 regulated by NF-kB pathway is one of the important signaling of inflammatory response in KC [27].

#### 3.2.7. Ketamine Induced Dysregulation of Autophagy and Inhibition of Angiogenesis

Autophagy acts a significant role in cell growth and development through the lysosomal mechanism to maintain cellular homeostasis. Mammalian target of rapamycin (mTOR) is involved in many signaling pathways to regulate cell growth, metabolism, apoptosis, and autophagy [64]. Dysregulation of mTOR has been demonstrated to be involved in a lot of diseases such as cancers, neurodegenerative diseases, diabetes, and genetic disorders [65]. Autophagy is induced by inactivation of mTOR and subsequently phosphorylation of class III phosphatidylinositol 3-kinase (PI3K-III) complex [40].

Lu et al. presented that ketamine and its metabolites increased the expression level of mTOR phosphorylation (8.0-fold) and significantly upregulated the level of Beclin1 (1.6-fold) and microtubule-associated protein light chain 3 (LC3) (1.45-fold) protein in the bladder of rats after ketamine treatment [40].

The expression level of serum vascular endothelial growth factor (VEGF) was significantly decreased in chronic ketamine abusers [66]. During angiogenesis, VEGFs interacting with VEGF receptors are essential angiogenic pathways for signaling networks, which involve the process of angiogenesis [67]. Lu et al. revealed that the presenting level of angiogenesis-related proteins was significantly decreased in the urothelial layer (UL) and muscular layer (ML), such as α-SMA, CD31, VEGF, VEGF-R1, VEGF-R2, laminin, and integrin-α6, of the rats bladder with KC compared to the control group [40].

The cellular signaling-related proteins of regulating autophagy in angiogenic remodeling include Erk1/2, p-Erk1/2, P38, p-P38, Akt, and p-Akt. In the UL of ketamine group, it has been shown that Erk1/2, p-Erk1/2, P38, p-P38, and Akt proteins significantly decreased, but not p-Akt protein [40].

All of these findings supported that ketamine initiated dysregulation of autophagy, inhibition of angiogenesis, and decreased the phosphorylation of Erk1/2 and p38, but up-regulated the phosphorylation of Akt. Therefore, the PI3K/Akt/mTOR pathway may be initiated by ketamine as a potential mechanism of the pathogenesis of KC [40].

### 3.3. In Vitro Study

#### 3.3.1. Increased Cytosolic Ca2^+^ Concentration May Be Lethal to the Urothelial Cells

Previous research has proposed that ketamine and its metabolite damage the bladder urothelium as direct toxic agents to result in KC [10,23]. Baker et al. discovered the relationship between cytosolic calcium concentration and direct toxicity of ketamine in KC [12]. The cytostatic concentration of more than 1 mmol/L of ketamine was cytotoxic, and it would produce a significant increase of cytosolic calcium concentration [12]. The persistent elevation of calcium concentration in cytosol due to ketamine treatment was associated with a significantly lower resting mitochondrial oxygen consumption rate (OCR) that resulted in pathological mitochondrial oxygen consumption and subsequently ATP deficiency. After depletion and run out of ATP in mitochondria, it can no longer maintain membrane potential and later form mitochondrial permeability transition pore. These results suggested that a prolonged, calcium-induced, intrinsic apoptotic pathway-driven response might damage the urothelial barrier in KC [12].

#### 3.3.2. Ketamine Induced Cytotoxicity and Apoptosis of Human Urothelial Cells

Using in vitro human urothelial cell lines assay including SV-HUC-1, RT4, and 5637, Shen et al. reported that cytotoxicity of ketamine treatment was dose-dependent and time-dependent. Ketamine stopped the bladder cell lines in the G1 phase cells and increased the level of sub-G1 population [25]. Ketamine decreased the barrier function of urothelial cell lines and subsequently increased the permeability of urothelial barrier [25].

Cytochrome c is a heme protein that is presented in the spaces between the inner and the outer membranes of mitochondria [68]. It is one component of the electron transport chain and transfers electrons between caspase IV and caspase III to cause caspase activation [69]. It functions as an important role to initiate the intrinsic pathway of cell apoptosis.

Simon et al. reported a significant average 2.4-fold increase in cytochrome c, which is an apoptotic mediator, of cytoplasmic fractions after 24 h of ketamine treatment (*p* < 0.05) [12]. This intrinsic apoptotic pathway may lead to extensive damage to urothelial cells. Furthermore, after treated with 3 mmol/L ketamine, it showed a rapid decrease in phosphorylation of Akt (serine 473) and ERK1/2 (threonine 202/tyrosine 204). Subsequently, it caused a decrease of the inhibition of GSK3β activity. It is inactive of GSK3β when phosphorylation occurred on the serine 9 of GSK3β. Treated with 3 mmol/L ketamine would reduce phosphorylation of the serine 9 on GSK3β, and consequently increased GSK3β activity that would contribute to increasing the activation of mitochondrial permeability transition pore. These results offer evidence of the direct toxicity of ketamine to the urothelial barrier by ketamine in KC.

All of these results provide the basic information for further examinations to investigate the mechanisms and effects of ketamine abuse in human with dysfunctional voiding.

## 4. Current and Potential Treatment for Ketamine-Related Cystitis

Because ketamine is the trigger of KC, the definitive treatment for KC is the cessation of ketamine abuse. Other developed and developing therapy will play the roles of relieving symptoms, improving quality of life, and helping recover the injured tissue. Among ketamine abusers who presented with LUTS, 51% reported improvement in urinary symptoms related to KC after cessation of ketamine used [5]. Most of the pathological changes of the urinary bladder are reversible, such as infiltration of mast cells or cellular apoptosis [70]. However, some pathological damages of KC are irreversible, such as collagen accumulation or contracture of the bladder wall [71]. 3.8% of ketamine abusers with KC, reported progressive urinary tract symptoms even after cessation of ketamine use [5]. An overview of therapeutic targeting strategies includes current medications used for KC and prospective therapies in the future (Figure 3).

KC shares many clinical and histological features with IC/BPS; most of the treatment strategies for KC were developed based on the treatments for IC/BPS [20]. Table 2 listed some of the clinical trials for KC:

### 4.1. Hyaluronic Acid

GAG covers the umbrella cells to protect the superficial layer of the bladder urothelium. GAG, which contains hyaluronic acid (HA), chondroitin sulfate, and heparan sulfate, forms a barrier of the urothelium with an anti-adherent activity from external invasion [75]. HA serves a vital role in cellular migration, proliferation, differentiation, and regulation of cell–cell interaction, which promotes tissue healing, tissue remodeling, immune response, and angiogenesis [76].

Intravesical instillation with HA has been used to treat KC based on the experience of treating IC/BPS [10,77,78]. Lai et al. treated 6 ketamine abusers with KC, who had ketamine abuse for 1.5 to 4.5 years, as cessation of ketamine along with intravesical HA instillation [72]. Of these patients, it showed improvement of the functional bladder capacities from 10–80 to 60–150 mL in 4 patients. However, there was no significant improvement in another 2 patients. Meng et al. administered intravesical instillation of HA with a dose of 40 mg in 50 mL of phosphate-buffered saline in five ketamine abusers, who failed oral medications previously, with a mean age of 22 ± 1.5 years and a mean duration of ketamine abuse as 68 ± 16.7 months [73]. HA was conducted weekly for 6 weeks and then followed by monthly for 3 months. The results reported significant improvement in VAS (from 7.0 ± 2.2 to 4.4 ± 0.6, *p* = 0.03), IPSS voiding subscore (from 16.2 ± 3.8 to 11.6 ± 4.2, *p* = 0.017), and O’Leary–Sant Interstitial Cystitis Symptom Index (ICSI) score (from 16.4 ± 2.7 to 13.6 ± 2.0, *p* = 0.016).

Lee et al. reported that intravesical HA treatment can improve the expression level of urothelial junction-associated proteins in mice with KC, including E-cadherin, claudin-4, ZO-1, and UPKIII. Additionally, HA treatment may reduce the presenting markers of inflammation and fibrotic biosynthesis, such as COX-2, TGF-β1, and type I collagen [32].

In mice, HA instillation to the bladder improved the production of HA receptors, including CD44, Toll-like receptor-4 (TLR-4), and receptor for HA-mediated motility (RHAMM). After the treatment, HA synthases type 1–3 increased, and hyaluronidases decreased in the urothelial cells of the bladder. Subsequently, HA improved the regeneration of urothelium in KC [32].

In summary, current evidence showed that intravesical HA treatment can regulate inflammatory responses, improve urothelial regeneration, and enhance bladder remodeling.

### 4.2. Botulinum Toxin A

Botulinum toxin A (BoNT-A) is a powerful neurotoxin that inhibits release of acetylcholine from nerve fibers and the urothelial cells to paralyze muscles [77]. Intravesical BoNT-A injection has been used to treat overactive bladder and IC/BPS, and effectively improved LUTS, such as frequency, urgency, pelvic pain, and increased bladder capacity [78,79]. BoNT-A had been demonstrated to reduce nerve growth factor production and ameliorate neurogenic inflammation in the IC/BPS patients [80].

Because of the similarity between IC/BPS and KC in clinical presentations and pathological characters, several studies examined the treatment effects of intravesical BoNT-A injection in KC. In a case series of thirty-six KC patients, intra-detrusor injection with 200 IU BoNT-A plus hydrodistention was performed in each patient. Lower urinary tract symptoms, including nocturia time, the interval of frequent urination, the void volume, the maximum urine flow rate, and bladder capacity, significantly improved at one-month treatment [74].

Lee et al. conducted intravesical instillation of liposomal onabotulinumtoxin-A (Lipotoxin) to treat female rats with KC. The study demonstrated that BoNT-A treatment could improve petechial hemorrhage, restore the urothelial barrier, and decrease bladder overactivity in rats with KC. The data also showed significantly increased expression of ZO-1 and E-cadherin and reduced production of substance *p* after Lipotoxin treatment. Additionally, BoNT-A treatment significantly inhibited inflammatory mediators (including IL-6, tumor necrosis factor-α (TNF-α), nuclear factor kappa B (NF-κB), and COX-2) in the detrusor muscles, and suppressed neuroreceptors in the bladder, such as mucosal transient receptor potential vanilloid 1 (TRPV1) and detrusor M2-mAChRs [31].

In summary, BoNT-A treatment is a potentially effective treatment for KC through mediating the neurogenic inflammation and suppressing the neuroreceptors in the bladder.

### 4.3. Bay K8644

Chen et al. reported that ketamine inhibits Cav1.2 of smooth muscles in the bladder and subsequently causes bladder dysfunction [15]. In mice, ketamine-induced pathology of the bladder was successfully abrogated upon intravesical instillation or intraperitoneal injection of Bay k8644, which is an agonist of Cav1.2. Bay k8644 can improve dysfunction of smooth muscles in KC and reverse functional and pathological changes of the bladder. Activation of Cav1.2 by Bay k8644 reduces urinary frequency and increases voiding volume. Bay k8644, the agonist of Cav1.2, may have the potential to rescue the pathological changes of the smooth muscles in KC.

### 4.4. Rapamycin

Rapamycin is an inhibitor of mTOR. Lu et al. presented that the expression level of autophagy-related proteins, including ATG12, ATG7, LC3, Beclin1, and VPS34, significantly increased in the rats treated with ketamine and rapamycin compared to the ketamine group [40]. However, the expression level of mTOR and p-mTOR protein significantly reduced in the ketamine and rapamycin group compared to the rats of the ketamine group. These results presented that rapamycin treatment could reduce the expression level of ketamine-induced mTOR protein and increase the expression level of autophagy-related proteins [40].

The expression level of angiogenesis-associated proteins was evidently inhibited in the UL and ML, which included α-SMA, CD31, laminin, VEGF, VEGF-R1, and VEGF-R2, of the rats treated with ketamine and rapamycin compared to the control rats. However, the inhibition was not as low as the rats of the ketamine group [40].

The study showed the expression level of p38 phosphorylation was increased in the ketamine and rapamycin group, but the expression level of p-Erk1/2 and p-Akt was decreased.

Lu et al. presented that rats with KC bladder treated with rapamycin could improve bladder function by inhibiting vascular angiogenesis, removing ketamine metabolites, eliminate eosinophil-mediated inflammation, and decreased bladder hyperactivity [40].

### 4.5. Wortmannin

Wortmannin is a potent PI3Ks inhibitor, which is derived from a fungal metabolite. Lu et al. proposed that the antiangiogenic effect of ketamine will prevent bladder repair. They have presented that the expression level of angiogenesis-related proteins (such as α-SMA, CD31, VEGF, VEGF-R2, laminin, integrin-α6) was markedly elevated in the UL of the ketamine and wortmannin group compared to the ketamine group, the ketamine and rapamycin group, and the control group of rats [40]. The study showed increased expression of Erk1/2, p-Erk1/2, p38, p-P38, Akt, and p-Akt proteins in the UL of ketamine and wortmannin group compared to the rats of the ketamine group.

Lu et al. demonstrated that wortmannin treatment in rats treated with ketamine could repair KC bladder by reversing antiangiogenic function by reducing basophil-mediated inflammatory response and increasing VEGF expression, and subsequently improving the angiogenesis of bladder [40].

### 4.6. Ba-Wei-Die-Huang-Wan (Hachimi-Jio-Gan)

Ba-Wei-Die-Huang-Wan (BWDHW), also known as Hachimi-jio-gan in Japan, is a traditional Chinese medicine for more than 1800 years and is currently used to treat diabetes and urinary frequency [81,82]. BWDHW is a herbal medicine with eight-species mixtures containing *Rehmannia glutinosa, Cornus officinalis, Dioscorea batatas, Alisma orientale, Porica cocos, Paeonia suffruticosa, Cinnamomum cassia,* and *Aconitum carmichaelii* [34]. In rat models of chemical cystitis, Tsai et al. reported that BWDHW could inhibit substance *p*, reduce infiltration of leukocytes, as well as suppress NF-κB and intercellular adhesion molecule-1 (ICAM-1) associated ROS-mediated injury to improve inflammation and hyperactivity of bladder [83]. In a cyclophosphamide-induced cystitis rat model which showed overactivity of bladder and acidic adenosine triphosphate (ATP) -induced bladder hyperactivity, BWDHW reduced the expression level of mucosal TRPV1, P2X2 and P2X3 purinergic receptors, as well as decreased M2- and M3-muscarinic receptors (mAChRs) in the bladder mucosa and detrusor muscles [35].

In a KC rat model, Lee et al. reported that ketamine and BWDHW treated rats exhibited decreased expression of substance *p*, less infiltration of monocyte and macrophage in suburothelial mucosa, and diminished deposition of interstitial fibrosis by reducing TGF-β1 expression compared to the rats treated with ketamine [34]. It also demonstrated that significantly lower expression of TRPV1, M2- and M3-mAChRs in the mucosal layer of the bladder and decreased levels of M2- and M3-mAChRs in the bladder detrusor muscle of BWDHW-treated rats. Furthermore, BWDHW treatment could decrease the expression of inflammatory mediators and fibrogenesis proteins, including IL-1β, IL-6, TNF-α, nuclear NF-κB, COX-2, ICAM-1, TGF-β1, collagen I, collagen III, and fibronectin.

In summary, BWDHW provided therapeutic effects in rats with KC by ameliorating the expression of neuroreceptors, inhibiting inflammatory mediators, and decreasing fibrogenesis to improve inflammation and overactivity of the bladder. It needs further studies to confirm the therapeutic effects of BWDHW in humans with KC.

According to the pathophysiological mechanisms of ketamine abused, many potential medicines may be used to treat KC, such as anti-inflammatory drugs, immunosuppressants, antioxidants, calcium channel blockers, and stem cell therapy. It needs further research and clinical trials to confirm the clinical availability and effects of these potential drugs.

## 5. Conclusions

Ketamine and its metabolites can cause disruption of the urothelial barrier by direct toxic damage and lead to sequential insults. The pathophysiology of KC involves inflammatory and immune reactions and multiple signaling pathways. No specific biomarkers help diagnose KC, although elevated IgE was found in KC patients and correlated with the severity of pain. It is a challenge to cure the disease if the patients keep using ketamine. Even though several treatments for KC have been reported, most of the suggested therapeutic strategies were palliative with minor effectiveness. There is currently no definite treatment. Overall, the clinical and basic research for KC is deficient. Further studies are needed to find out molecular targets that help to restore the urothelial barrier and prevent further damage of the chronic inflammation process.

## Figures and Tables

**Figure 1 biology-11-00502-f001:**
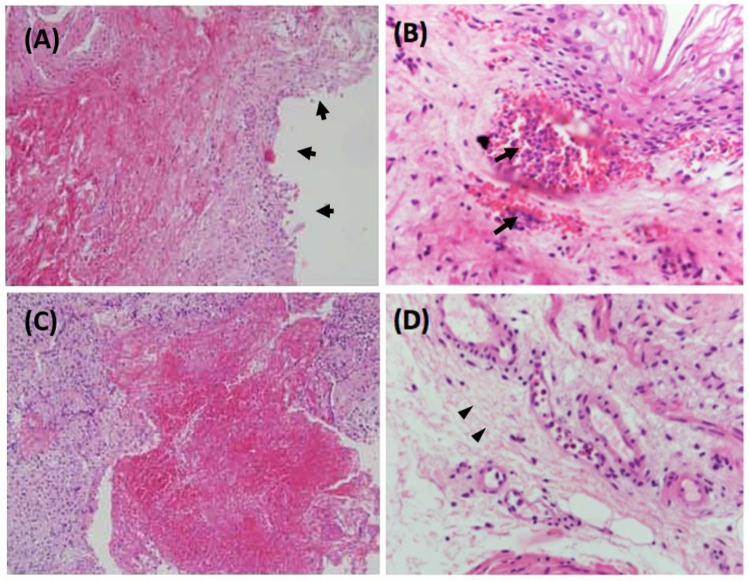
Histology of the urinary bladder in ketamine-induced cystitis (KC) patients stained with hematoxylin and eosin showed (**A**) denuded bladder mucosa (short arrows), (**B**) infiltration of neutrophils and eosinophils in the bladder stroma, and accumulation of intravascular eosinophils (arrows), (**C**) ulceration of the urothelium and (**D**) collagen deposition in the submucosal layer (arrowheads). (**A**,**C**) at 200× that were modified from [17]; (**B**,**D**) at 400× that were modified from [18].

**Figure 2 biology-11-00502-f002:**
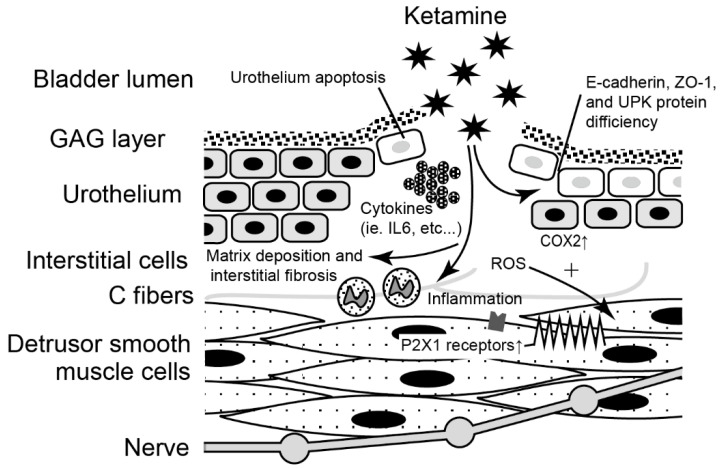
Schematic of possible mechanisms of ketamine-related cystitis, contains activation of inflammatory cells, dysfunction of bladder-urothelial barrier, dysregulation of neurotransmission, cell apoptosis, and oxidative stress. ROS: reactive oxygen species, ZO-1: zonula occludens-1, UPK: uroplakins, COX2: cyclooxygenase-2.

**Figure 3 biology-11-00502-f003:**
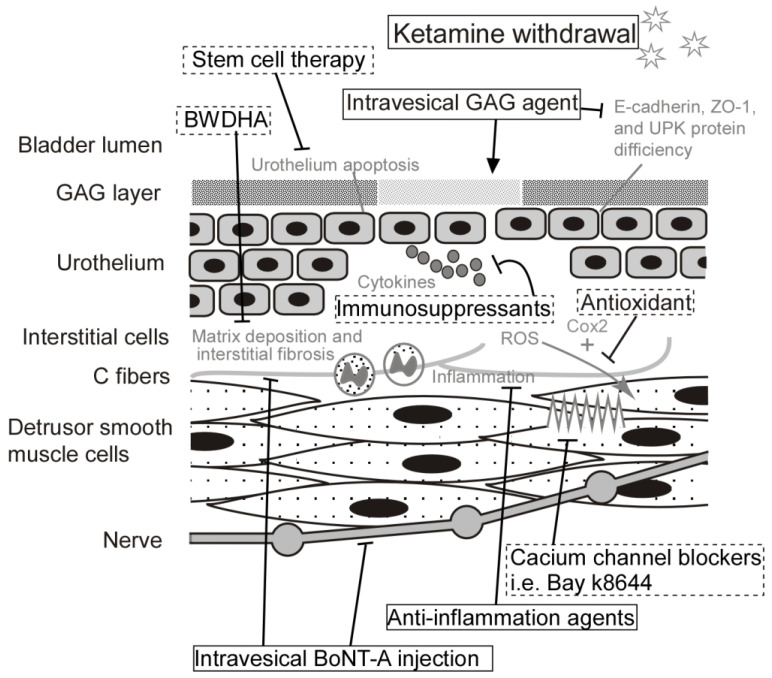
Schematic of therapeutic targeting strategies contains current medications already used for KC patients and potential therapies need further studies in the future (dash-line boxes). KC: ketamine-induced cystitis, BWDHA: Ba-Wei-Die-Huang-Wan, GAG: glycosaminoglycan, ZO-1: Zonula occludens-1, UPK: Uroplakin, ROS: reactive oxygen species, COX2: cyclooxygenase-2, BoNT-A: botulinum toxin-A.

**Table 1 biology-11-00502-t001:** Possible molecular etiology and potential targeted treatment of KC based on human, animal, and in vitro studies.

Molecular Etiology	Human Study	Animal Study	In Vitro Study	Potential Treatment
Inflammation	Hypersensitivity reaction due to ketamine or its metabolites [28] Ketamine causes releasing of cytokines (i.e., IL6) as well as imbalance of immune response [18]	Increased cytokine expressions, such as IL-1β, IL-6, CCL-2, CXCL, CXCL-10, NGF, and COX-2 [29,30]		Anti-inflammatory drugs Immunosuppressants Intravesical BoNT-A injection [31]
Urothelial junction-associated protein	Decreased expression of E-cadherin in the urothelial cells of KC bladder [24]	Decreased level of GAG, E-cadherin, ZO-1, and urothelial umbrella cells [22,32]		Intravesical instillation of GAG agents, such as Hyaluronic acid [32]
Ion channels in the bladder mucosa	Higher presenting level of TRPV1 and TRPV4 in the bladder mucosa of KC bladder [33]			Intravesical BoNT-A injection [31] BWDHA [34,35]
Oxidative stress		Increased expression of ROS [26] Significantly decreased in Mn-SOD (SOD2), Cu/Zn-SOD (SOD1) [26]		Antioxidant
Neurotransmission alternation		Increased expression of P2X1 purinergic receptors [36] Possible M2 and M3 muscarinic receptors [34,36]		Intravesical BoNT-A injection [31] Anticholinergics [34,36] BWDHA [34,35]
Ion channels in the bladder smooth muscle		Inhibition and decreased expression of Cav1.2 [15] Downregulation of KCNMA1 and KCNMB4 genes that involve in calcium signaling pathway [37,38]		Agonist of Cav1.2 (Bay k8644) [15]
Fibrosis-related genes		Upregulation of COL I, COL III, fibronectin, and TGF-β [39]		BWDHA [34]
Keratin family genes		Downregulation of keratin 6 a, 13, 14 [25]		
Other signal pathways		Matrisome (ECM glycoproteins, matrisome and matrisome associated) Calcium signaling pathway Small cell lung cancer MAPK signaling Regulation of actin cytoskeleton Neuroactive ligand receptor interaction Complement and coagulation cascades [37]		
ECM related genes		Upregulation of FN1, fibulin 2, fibrinogen-like 2, LAMC2, COL1A2, VCAN, AGT and C-type lectin domain family 4 member D [37]		
Autophagy and angiogenesis		Ketamine induced dysregulation of autophagy and inhibition of angiogenesis (ketamine triggered PI3K/Akt/mTOR pathway) [40]		Rapamycin [40] Wortmannin [40]
Cytosolic calcium concentration			Increased level of cytosolic calcium concentration [12]	Calcium channel blockers
Cell apoptosis			Ketamine stopped the urothelial cell lines in G1 phase [25] Increased expression of cytochrome c, caspase IV and III [12]	Stem cell therapy

NGF, nerve growth factor; COX-2, cyclooxygenase-2; HA, hyaluronic acid; BoNT-A, botulinum toxin A; TRPV, transient receptor potential cation channel subfamily V; BWDHA, Ba-Wei-Die-Huang-Wan; BDNF, brain-derived neurotrophic factor; IgE, immunoglobulin E; ZO-1, zonula occludnes-1; ROS, reactive oxygen species; Cav1.2, L-type calcium channel; KCNMA1, potassium calcium-activated channel subfamily Mα1; KCNMB4, potassium calcium-activated channel subfamily M regulatory β subunit 4; COL I, collagen I; COL III, collagen III; TGF-β, transforming growth factor-β; ECM, extracellular matrix; MAPK, mitogen-activated protein kinase; FN1, fibronectin 1; LAMC2, laminin γ2; COL1A2, collagen type 1 α 2; VCAN, versican; AGT, angiotensinogen; PI3K, phosphatidylinositol 3-kinase; mTOR, mammalian target of rapamycin.

**Table 2 biology-11-00502-t002:** Current therapies used for KC patients and their outcomes.

	Intravesical HA Instillation	Intravesical BoNT-A Injection
	Lai, Y [72] Meng, E [73]	Zeng, J [74]
Study design	Case series	Prospective study
Numbers of ketamine abusers	6 (4 men, 2 women) 5 (1 men, 4 women)	36 (30 men, 6 women)
Age, years	24.8 (21–30) 22 ± 1.5 (21–25)	26.0 (19–38)
Duration of ketamine abuse, months	37.2 (18–54) 68 ± 16.7 (48–84)	12–60
Drug administration	Unknown 40 mg of HA in a 50 mL of phosphate-buffered saline was instillation once weekly for 6 weeks and then once monthly for 3 months (totally 9 instillations)	200 U (injected into the bladder walls at 40 sites) followed by cystoscopic hydrodistention under a pressure of 80 cm and maintained the bladder capacity at 150–200 mL for 5 min
Outcomes	66.7% (4/6) patients improved, FBC↑ 66.7% (4/6) improved	1 month after BoNT-A treatment: nocturia↓, interval between micturition↑, void volume ↑, maximum flow rate ↑, bladder capacity↑, ICSI score ↓, and ICPI score ↓
	At 1 month after intravesical instillation of HA: VAS ↓ IPSS voiding subscore ↓ ICSI scores↓	

KC, ketamine-related cystitis; HA, hyaluronic acid; BoNT-A, botulinum toxin A; FBC, functional bladder capacity; VAS, visual analog scale; IPSS, international prostate symptom score; ICSI, O’Leary–Sant interstitial cystitis symptom index; ICPI, O’Leary–Sant interstitial cystitis problem index.

## Data Availability

No new data were created in this study. Data sharing does not apply to this article.

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
