# Peer review of "Molecular Pathophysiology and Potential Therapeutic Strategies of Ketamine-Related Cystitis"

_biology, 2022, doi:10.3390/biology11040502_

Round 1

Reviewer 1 Report

Manuscript ID: biology-1617489

Type of manuscript: Review
Title: Molecular Pathophysiology and Potential Therapeutic Strategies of Ketamine-related Cystitis

In this manuscript, the authors give an overview of what has been published on the use of ketamine, the pathologies inherent to its misuse, with special emphasis on cystitis, the search for reliable biomarkers and, finally, treatments that may improve the condition of individuals.

All data presented were based on publications resulting from clinical and basic research.

With the presentation of tables, it is much easier for those who read the manuscript to find the publication on a given subject.

I think this article will make contribution to the literature. This manuscript can be accepted and published, in my opinion.

Author Response

Thank you for your comment and accepting this article to publish. 

Reviewer 2 Report

Chen et al., have submitted the review entitled “Molecular Pathophysiology and Potential Therapeutic Strategies of Ketamine-related Cystitis”.

This is an excellent review. In this review, the authors have comprehensively described the abuse of ketamine and associated diseases. Particularly, the authors have focused on the molecular pathophysiology and therapeutic avenues for ketamine-related cystitis. In addition, the authors have made worthy discussion by using animal models and in vitro studies.

Is figure 1 is original or copyrighted from the other sources? This should be mentioned in the figure legend.

It would be attractive to illustrate a figure with therapeutic targeting strategies.

Author Response

Thank you for your valuable comment and suggestions which are very helpful for our manuscript.

This manuscript is a resubmission of an earlier submission. The following is a list of the peer review reports and author responses from that submission.

Round 1

Reviewer 1 Report

The article by Chen and Meng “Molecular Pathophysiology and Potential Therapeutic Strategies of Ketamine-related Cystitis” describe the pathophysiology of

 ketamine-related cystitis. Given that recreational drug use has become a growing concern, the article is a laudable approach to discuss the pathologic and other side effects of uncontrolled ketamine use. The review article provides theoretical bases for developing in vitro experiments, animal models, and clinical trials. Further, this review summarized evidence of molecular mechanisms of KC and potential treatment strategies for KC.

There are some minor comments and concerns

  1. Table 1 is difficult to read.
  2. There are many instances that neuronal cells and their network control the tissue function and homeostasis. Given the involvement of muscle contraction and Ketamine-related Cystitis, the authors can discuss the participation of neuronal networks.
  3. Sections 1.C & 3.2.5 need more discussion.
  4. Is there any involvement of mucins since the mucus layer is damaged mainly because of ketamine?

Reviewer 2 Report

The manuscript ‘Molecular Pathophysiology and Potential Therapeutic Strategies of Ketamine-related Cystitis’ ..’focuses on current evidence of molecular pathophysiology and possible therapies for Ketamine Cystitis’. And the introduction states: ‘How ketamine affects the bladder and causes the damage of urothelial cells remains unclear.’ I agree with that statement. However, when I go through the article and finally read the conclusion, my disappointment is great. I (read this as a clinician and) don't think the conclusion adds much to the introduction. The summary of preclinical and animal lab-research is impressive and, I do not think much is missing. However, the conclusion is that: ‘research help to elucidate the molecular mechanism of KC and identifying the specific molecular targets for effective therapies’ and that (in short): there is yet no good (addiction) management nor curative treatment available and finally that: ‘further studies are required’. I give one clear example of a sub-conclusion of one of the paragraphs: L373: In summary, BoNT-A treatment is a potentially effective treatment for KC through mediating the neurogenic inflammation and suppressing the neuroreceptors in the bladder. Ref 73 is not at all a valid study to suggest this (apart from ‘potentially’), and as is a problem in all animal studies: (more) frequent voiding can not be ‘translated’ into detrusor overactivity (or the human! clinical symptoms syndrome (syndrome!) of overactive bladder syndrome, when it can not be assessed whether the animals void with higher frequency because of (inflammation and) pain. Frequent voiding because of pain (and or inflammation) would not be ‘overactive bladder syndrome (or ‘idiopathic’ detrusor overactivity (not ‘hyperactivity’ nor ‘instability’) in the clinic(-al care). I ask to scrutinize -and adapt- all statements with this in mind. The generalizability of animal lab studies to a clinical syndrome is (extremely) low. The statement in L 119 ‘hypersensitivity reaction’ due to interact with the urothelial cells.. should be clarified: hypersensitivity is not introduced nor explained in ref 25 and used frequently in the manuscript. It should (I assume) clearly state (something alike) ‘immunological hyperreactivity’ to separate from (clinical) hypersensitivity (and or pain). -also, because what is mentioned about clinical pain with frequent voiding. Furthermore, without further evidence given, IgE is -in my view- likely to be much too unspecific to be useful in clinical (early?) diagnosis.

L 165 sub-concludes: ‘Disruption of the urothelial barrier can lead to leakage of urinary potassium or irritative agents to the suburothelium and cause an inflammatory reaction and provoke cystitis.’ What I would appreciate is that the conclusion includes the -based on all reviewed research- hypothesis (of the authors) regarding the most likely sequence of events leading to pathophysiology. Does the problem start with urine (metabolites) affecting the urothelium and so-forth…. Or are hematological -&interstitial fluid- effects also relevant. For the last: neurological mechanisms are frequently mentioned (nerve damage). Is nerve damage caused because of ‘urothelial defects/leakage’ of because of ‘hematological toxicity’. Sure, this is not precisely answered but a meticulous review of the literature (as is done) should lead to a much more precise, applicable, and sensible conclusion than given now. That (including considerations about bench (animal-lab) -to -bedside translation) would help to direct (further) research. I ask to reduce all overstatements regarding the applicability of lab-findings regarding treatment (and maybe to include lab-research that shows that stopping Ketamine allows the lower urinary tract to (histologically) recover, or the lack thereof, when the most important clinical conclusion is that advise to the sufferers of KC.) the study is concentration on urothelium, I consider it (also regarding the sequence of events as mentioned here above) relevant to also include sub-urothelial but more important detrusor (interstitial) histo(pathology) cell receptors and or transmitters, unless there is convincing evidence that this is not at all relevant for management or treatment. If that is not considered relevant I ask to be very careful with a comparison with interstitial cystitis (unless there is evidence enough to say that the (detrusor) interstitium is the common pathophysiology in both) and to abstain from comparing with bladder pain syndrome.

Round 2

Reviewer 2 Report

The manuscript: ‘Molecular Pathophysiology and Potential Therapeutic Strategies of Ketamine-related Cystitis” is somewhat adapted as a reaction on the reviewers comments. But the conclusion is still not a reflection of the study and the study still contains significant overstatements about treatments (treatments that hardly match the pathophysiology (as is reviewed)). I can agree with the statement that damage to the urothelium by ketamine (metabolites) is the beginning of all the problems although this is not firmly proven (especially not in ref 27 (see L27)). That ketamine ‘can cause’ (citing from the conclusion) may be true, but that it does, is ‘only’ the most likely clinical hypothesis. That research ‘will help’ (citation again) is true. Can the authors say how their summary of all studies has helped? (and put especially this in the conclusion). If a biomarker (IgE) associates with a symptom, the symptom is easier (and cheaper) to obtain than the biomarker. Therefore use pain a the marker. I agree: there is no treatment. And I think that the summary of all research shows that there is hardly a mechanism of action for anticholinergics or antimuscarinics as there is also no mechanism of action for botulinum toxin. Every molecule is a ‘potential therapy’ but that is not how current (pharmaco-) therapy is developing. It can be made clear that these treatment cannot have any effect on the causative lesion (that the authors postulate) and at best have some symptomatic (palliative) effect. Yes the last sentence is true, but the title of the manuscript suggests much, much more. Please summarize what you have discovered (through your reading of other’s work) about the pathophysiology and potential key features –accessible for (pharmaco_)treatment and conclude about that (ony). Please also think about what you know that are dead-end streets in (lab- and (pre)clinical) research.

Round 3

Reviewer 2 Report

The manuscript: ‘Molecular Pathophysiology and Potential Therapeutic Strategies of Ketamine-related Cystitis’ is revised again, but the core problem is still not solved.

You have stated that the primary problem of KC is that the urothelial barrier is damaged and that the suburothelial layer suffers from that (and responds immunologically). I can accept this (after you explicitly stated that you did not find evidence for ‘direct’ systemic effect on e.g. detrusor or surrounding structures (nerves vessels)).  Then you conclude that anticholinergics ‘need further study’. There is in my view not any reason to do further study with  anticholinergics and or antimuscarinics, there is no known mechanism of action of anticholinergics/antimuscarinics to repair urothelial and or suburothelial layer(s). Therefore: anticholinergics and or antimuscarinergics can (at best) only be ‘palliative’. And certainly not ‘curative’. I will not accept that you not state this explicitly. Unless you have evidence that anticholinergically or antimuscarinergically acting medications are repairing the cause of the problem (as your title is ‘molecular pathophysiology of KC’ and potential therapeutic…). You explain the pathophysiology and subsequently suggest treatments without any (reported by you) mechanism of action towards this pathophysiology. You cannot do this. The title of your manuscript requires that you associate the pathophysiology with ‘therapeutic strategy’ and not that you summarize a pathophysiology and subsequently report strategies that have no relation to the pathophysiology at all.

The same is for BTx: unless you can show (evidence) that BTx has any direct effect on the pathophysiology that you have postulated on the basis of the evidence that you have summarized, you cannot say that ‘intravesical’ (line 390 should be 'intra-detrusor plus hydrodistension') BTx is a potential curative treatment. And I think that ‘injection’ in L 458 should not be there if you want to say something about intravesical instillation of liposomal BTx (but I think that this (liposomal BTx) should be concluded in the paragraph (4.3.) only, because this is not at all clinically (specifically) established at present). The overall conclusion should say that your summary has found that the urothelial barrier etc. etc….. And also that most of the (pharmaco)therapeutic strategies are not very effective (apart from stopping K) and some of the treatments are palliative (with also little effect). I would say: Further studies are needed to find ‘molecules’ that protect or help restoring the urothelial barrier (or maybe: ‘that modulate the secondary responses’), further studies for palliation are not primary needed (although also that may also be a goal of research but that should be mentioned specifically: palliative treatments do not act on the ‘molecular pathophysiology’ (and do not act as a  therapeutic strategy).
